# An Interleukin-4 and Interleukin-13 Induced Atopic Dermatitis Human Skin Equivalent Model by a Skin-On-A-Chip

**DOI:** 10.3390/ijms23042116

**Published:** 2022-02-14

**Authors:** Kyunghee Kim, Hyeju Kim, Gun Yong Sung

**Affiliations:** 1Interdisciplinary Program of Nano-Medical Device Engineering, Hallym University, Chuncheon 24252, Korea; seoulhee92@naver.com; 2Major in Materials Science and Engineering, Hallym University, Chuncheon 24252, Korea; heajoo1450@naver.com; 3Integrative Materials Research Institute, Hallym University, Chuncheon 24252, Korea

**Keywords:** atopic dermatitis, skin-on-a-chip, human skin equivalent, interleukin-4, interleukin-13, carbonic anhydrase II

## Abstract

Currently, the mechanism of progression of atopic dermatitis (AD) is not well understood because there is no physiologically appropriate disease model in terms of disease complexity and multifactoriality. Type 2 inflammation, mediated by interleukin (IL)-4 and IL-13, plays an important role in AD. In this study, full-thickness human skin equivalents consisting of human-derived cells were fabricated from pumpless microfluidic chips and stimulated with IL-4 and IL-13. The morphological properties, gene expression, cytokine secretion and protein expression of the stimulated human skin equivalent (HSE) epidermis were investigated. The results showed epidermal and spongy formations similar to those observed in lesions in AD, and decreased expression of barrier-related filaggrin, loricrin and involucrin genes and proteins induced by IL-4Rα signaling. In addition, we induced the expression of *carbonic anhydrase II (CAII)*, a gene specifically expressed in the epidermis of patients with AD. Thus, AD human skin equivalents can be used to mimic the key pathological features of atopic dermatitis, overcoming the limitations of existing studies that rely solely on mouse models and have been unable to translate their effects to humans. Our results will be useful for future research on the development of therapeutic agents for atopic dermatitis.

## 1. Introduction

Atopic dermatitis (AD) is a chronic inflammatory skin disease associated with widespread barrier dysfunction and T helper 2 (Th2) cytokines [1,2]. AD is an imbalance in the Th2 immune response where there is an increase in the gene expression levels of major Th2 cytokines during the acute phase [3,4]. It is characterized by dry erythema and severe itching. At the histological level, acute AD exhibits spongiosis in the basal epidermal layer [5]. Spongiosis is a morphology where the intercellular space of the skin tissue expands as edema liquid accumulates due to changes in cohesion between skin normal human epidermal keratinocytes (NHEKs). Atopic dermatitis in the chronic phase, lichenificated lesions, typically shows epidermal hyperplasia and parakeratotic hyperkeratosis. Stratum basal expansion, stratum spinosum and stratum granulosum (SG) reduction and stratum corneum (SC) thickening are observed because of terminal keratin cell differentiation defects [6,7,8]. Currently, the etiopathogenesis of AD is not well understood, at least partially because there is no physiologically appropriate disease model in terms of disease complexity and multifactorial nature. AD is likely caused by epidermal barrier alteration and Th2 immune response dysregulation [9,10].

The formation of skin barriers and the effects of cytokines on selected targets during differentiation processes in cell culture and animal studies have been studied [11]. In the stratum basale layer, cytokines including IL-4, IL-13, IL-20, IL-24, IL-22, IL-1α and tumor necrosis factor inhibit keratin synthesis, which affects profilaggrin synthesis in the next layer. These cytokines suppress the formation of the filaggrin/keratin network and weaken the function of the barrier. In the SC layer, desquamation is promoted by IL-4 and IL-13. During the differentiation process, IL-4, IL-13 and IL-22 suppress the expression of involucrin and loricrin and the formation of the cornified envelope [11].

In skin lesions in patients with AD, the Th2 cytokines IL-4 and IL-13 are expressed at much higher levels. Barrier dysfunction in AD is associated with downregulation of barrier-related molecules such as filaggrin (FLG), loricrin (LOR) and involucrin (IVL) [12]; Figure 1 shows a schematic diagram summarizing this.

Recently, regulation of animal testing has increased interest in alternative models [16], which either do not use animals or reduce the number of animals or reduce pain (3Rs; replacement, reduction and refinement) [17]. Existing animal models are expensive and bear ethical concerns. In addition, the structure of the skin and the immune response in animals differ from those of humans [18]. To overcome this limitation, three-dimensional (3D) cell culture models for culturing human-derived cells using extracellular matrix (ECM) scaffolds are being investigated. Three-dimensional culture has the advantage of creating an environment that is similar to complex in vivo conditions [19,20]. Treatment of AD currently requires a viable in vitro model to test new molecules [21].

Skin-on-a-chip (SOC), a microphysiological system technology in 3D culture, utilizes human skin equivalent (HSE), which has the same characteristics as human skin, as it allows drug screening to be performed quickly and easily [22,23]. A pumpless SOC can be operated more simply and can be tested on drugs using a microphysiological system [24,25]. This system can create a biomimetic environment that reproduces various physiological functions, such as the supply of nutrients to cells via microfluidic channels and the removal of cell waste without a pump.

The aim of this study was to develop full-thickness HSE from a pumpless SOC stimulated with IL-4 and IL-13 to induce AD, to investigate the morphological properties and to evaluate the induction of AD by our skin substitutes and their potential as a drug test evaluation model for therapeutic agents for AD.

## 2. Results

### 2.1. Alteration of Tissue Morphology of the AD-like HSE Model

Morphological changes in the AD-like HSE model were examined by hematoxylin and eosin (H&E) staining (Figure 2a–l). From the results of H&E staining, five regions were randomly designated, and the total thickness of epidermis and the thickness of SC only were measured and quantified (Figure 2m,n).

As shown in Figure 2m,n, the EDL thickness on days 3, 7 and 14 under the 0 ng/mL IL-4/IL-13 (control) condition was 22.20 to 29.22 μm. The SC layer was shown to gradually increase with the culture period (0 to 11.34 μm). Among the controls cultured for 7 days, the total thickness of EDL was measured to be 29.22 μm on average, and the SC layer was measured to be 8.83 μm on average. Although the EDL except for the SC was maintained at a level similar to that of 3 days of culture, it can be considered as the most well formed tissue because the SC layer was formed properly. With 7 days of culture at 0 ng/mL IL-4/IL-13, the SC was well formed (Figure 2a–c). With 3 days of treatment, there was insufficient time for differentiation, and the plurality of layers constituting the epidermal layer (EDL) was not clearly formed; in particular, the SC layer was not clearly differentiated (Figure 2a). With H&E staining, the HSE control cultured for 7 days showed the most distinctive differentiation and thickness of various layers of EDL (Figure 2b,m,n).

Samples treated with 15 ng/mL IL-4/IL-13 for 3 days showed poorly differentiated EDL, along with severe intercellular spongiosis and a tremendous amount of intercellular spaces. Due to the poorly differentiated EDL, SC was not formed and only the thickness of the unstable EDL increased (Figure 2j,m,n). The higher the concentration of IL-4/IL-13 treatment for 7 days, the thicker the EDL formed (Figure 2e,h,k). Compared with the control group, IL-4/IL-13 treatment yielded very thick EDL, even hyperkeratosis. More intercellular space was formed by AD-HSE stimulated for 7 days than for 3 days (Figure 2d,g,j). The thickness after 7 days was measured to be approximately 29.22 μm at 0 ng/mL, 31.62 μm at 5 ng/mL, 52.14 μm at 10 ng/mL and 48.40 μm at 15 ng/mL. Compared to the control group, a significant increase in EDL thickness was observed in the experimental groups treated with 10 and 15 ng/mL. Proper differentiation of each layer of EDL did not occur; however, there was acanthosis in which the stratum spinosum layer formed was abnormally thick (Figure 2e,h,k,m). Figure 2c shows the control cultured for 14 days, in which SC formation is evident, and the remaining layers form a thinned EDL. Figure 2f,i,l shows the increase in the thickness of all layers, including SC, when treated with IL-4/IL-13. The SC layer is also scattered without being firmly connected. In the 15 ng/mL group, SC could not adhere and showed a scattered morphology (Figure 2l). The degree of skin lesions depended on the duration of stimulation, with the maximum effect achieved with 7 days of stimulation (Figure 2k).

### 2.2. Downregulation of Barrier Function-Related Proteins in AD-HSE Stimulated with IL-4 and IL-13

Immunohistochemistry (IHC) staining was performed to investigate the expression of barrier function-related proteins (Figure 3). LOR is expressed in the SG of the epidermis. As shown in Figure 3d, there was no obvious change in LOR expression by treatment with IL-4/IL-13 for 3 days, but the test group treated for 7 and 14 days showed less LOR staining than the control group. On 7 days, IL-4/IL-13-treated AD-HSE showed LOR expression 0.75 to 0.49 times that of the control group, and a significant decrease in LOR expression was observed in AD-HSE treated with 15 ng/mL IL-4/IL-13 (*p* < 0.01). On day 14, the IL-4/IL-13 treatment group showed LOR expression 0.85 to 0.54 times that of the control group. There was significant inhibition of expression at 5 ng/mL (*p* < 0.05) and 10 ng/mL (*p* < 0.01).

IVL, expressed in the SS layer, showed a decrease in AD-HSE induced by IL-4/IL-13. The expression of IVL was 1 to 0.54 times that of the control group for AD-HSE on 3 days, 0.25 to 0.06 times that of the control group for AD-HSE on 7 days and 0.54 to 0.85 times that of the control group for AD-HSE on 14 days. The higher the IL-4/IL-13 concentration for 7 and 14 days, the greater the decrease in protein expression (Figure 3e). There was a significant decrease in the 15 ng/mL IL-4/IL-13 treatment group for 3 days (*p* < 0.01) and 7 days (*p* < 0.0001). There was also a significant decrease in 14 days treatment group (*p* < 0.001, 0.0001).

FLG is expressed in the SG and at the interface between the SG and SC. In the control group, FLG showed increased protein expression with longer epidermal differentiation. However, IL-4/IL-13-induced AD-HSE showed reduced FLG expression. In AD-HSE induced for 7 days, the expression was 0.92–0.12 times that of the control group, and there was a significant decrease (*p* < 0.0001) in the 15 ng/mL. AD-HSE induced for 14 days showed a significant expression (*p* < 0.0001) of 0.41–0.07 times that of the control group. In particular, there was a large decrease in its expression with 15 ng/mL IL-4/IL-13 (Figure 3f).

To confirm that skin damage and psoriasis were induced in our AD-HSE, we examined KRT10 and KRT16 expression. The expression of KRT10 protein increased as the period of induction of epidermal differentiation in the control group increased. There was a significant decrease (*p* < 0.001) of 0.05 times compared to the control group in 10 ng/mL AD-HSE induced for 3 days. AD-HSE induced for 7 days showed a significantly reduced (*p* < 0.0001) expression (expression of 0.11–0.22 times that of the control group). The expression of KRT10 was decreased in AD-HSE induced by IL-14 and IL-13, and the greatest decrease in expression was observed in AD-HSE induced for 7 days (Figure 3g). KRT16 protein, in contrast, did not show a significant difference in expression between the control and IL-14- and IL-13-induced AD-HSE (Figure 3h).

### 2.3. Alteration of Epidermal Morphologies in AD-HSE Stimulated with IL-4 and IL-13

IL-4 and IL-13 stimulation weakens the epidermal barrier and changes the epidermal morphology. We thus analyzed the epidermal morphology of AD-HSE by SEM. Figure 4 shows the 500 magnification result of the SEM. The cross-sectional view of AD-HSE confirmed the proper formation of the stratum corneum. The SC stratum of cytokine-treated AD-HSE was in an unstable form. When compared with control, unevenness and rough morphology of the epidermal topography were commonly observed in AD-HSE treated with IL-4/IL-13.

### 2.4. AD-Related Gene Expression in AD-HSE Stimulated with IL-4 and IL-13

Treatment of AD-HSE with IL-4 and IL-13 was shown to reduce *FLG* and *IVL* gene expression in all experimental groups except those stimulated at 10 ng/mL for 14 days. In particular, the expression of *FLG* (*p* < 0.05) and *IVL* genes (*p* < 0.01) was most significantly suppressed in AD-HSE stimulated with IL-4/IL-13 at 15 ng/mL (Figure 5a,c). The expression of FLG decreased in AD-HSE stimulated for 3 days was 0.28-fold (5 ng/mL), 0.26-fold (10 ng/mL) and 0.14-fold (15 ng/mL) of the control group. In the AD-HSE stimulated for 7 days the expression of FLG was significantly decreased 0.10-fold of the control group (*p* < 0.01) (Figure 5a). Expression of LOR decreased when stimulated with IL-4/IL-13 for 3 days, but increased when stimulated with 10 ng/mL for 7 and 14 days (Figure 5b). In AD-HSE stimulated for 7 days, the expression of IVL decreased was 0.48-fold (5 ng/mL), 0.56-fold (10 ng/mL) and 0.12-fold (15 ng/mL) of the control group. Significance was shown at *p* < 0.01, 0.05 and 0.0001, respectively (Figure 5c). At 15 ng/mL, the expression of *FLG*, *IVL* and *LOR* genes was significantly reduced. This is consistent with previously reported in vitro results for IL-4 and IL-13 [26,27,28,29]. As shown in Figure 5d, we found that IL-4 and IL-13 strongly increased *CAII* gene expression. In AD-HSE stimulated for 3 days, the expression of *CAII* was 2.92-fold (5 ng/mL), 13.69-fold (10 ng/mL) and 17.30-fold (15 ng/mL) that of the control group. There was a significant difference (*p* < 0.001, 0.0001) from the control at 10 and 15 ng/mL. In AD-HSE stimulated for 7 days, the expression of *CAII* was 9.68-fold (10 ng/mL) and 11.77-fold (15 ng/mL) that of the control group. There was a significant difference (*p* < 0.0001) from the control at 10 and 15 ng/mL on 7 days (Figure 5d).

### 2.5. Secreted Cytokines in AD-HSE Stimulated with IL-4 and IL-13

Figure 6a shows the ELISA results of AD-HSE stimulated with 15 ng/mL of IL-4/IL-13 for 3, 7 and 14 days. The expression levels of IL-4/IL-13 were highest in AD-HSE stimulated for 14 days and lowest in AD-HSE stimulated for 7 days. Treatment was performed at the same concentration, but different numerical values were expressed depending on the treatment period. Cytokine expression was observed to change most when AD-HSE was cultured for 14 days. Several concentrations of IL-4/IL-13 were used for 14 days to assess the response of other cytokines (Figure 6b). HSE not treated with the IL-4/IL-13 had the lowest expression of most cytokines. GCSF and stem cell factor (SCF) were the least expressed at 5 ng/mL IL-4/IL-13, and plasminogen activator inhibitor-1 (PAI-1) was the lowest at 10 ng/mL IL-4/IL-13. Figure 6c displays a comparison of the experimental groups in Figure 6a,b with a heat map. TNF-α, IFN-γ, GM-CSF, IL-1α, IL-8, interferon-inducible protein 10 (IP-10), RANTES, vascular endothelial growth factor (VEGF), epidermal growth factor (EGF), resistin, IL-12, IL-13, platelet-derived growth factor BB (PDGF-BB), placental growth factor-1 (PIGF-1), beta-nerve growth factor (β-NGF), monocyte chemoattractant protein-1 (MCP-1), macrophage inflammatory protein-1α (MIP-1α), IL-4, IL-10, basic fibroblast growth factor gene (FGFb), leptin, insulin-like growth factor I (IGF-1), transforming growth factor-β (TGF-β), Adipo, IL-17A and IL-1β were upregulated in IL-4/IL-13-induced AD-HSE. In the case of IL-2 and IL-6, the ELISA results exceeded the linear range of the kit, so we were unable to accurately compare the expression of these two cytokines.

## 3. Discussion

The purpose of this study was to develop a disease model that exhibits morphological and molecular characteristics of AD using HSE stimulated with specific cytokines, known as the cause of AD, using a microphysiological system. To develop an AD model that exhibits appropriate molecular characteristics, the Th2 cytokines IL-4 and IL-13 were used to induce the AD phenotype [30,31], as they are expressed at very high levels in skin lesions in AD patients. This is because the immune response in the acute phase of the disease is predominantly regulated by Th2 cells [32].

Since the levels of IL-4/IL-13 in AD skin are unknown, we tested various levels of cytokines added to HSE. Cytokine concentrations were chosen according to those used in other in vitro studies. Due to the small size of the microphysiological system used in this study, it was important to find suitable levels of cytokine levels and appropriate stimulation methods for the HSE scale. At cytokine concentrations of 15 ng/mL and above, AD-related reactions were more common, but the reactions were less efficient than those at higher concentrations. Therefore, we selected the concentrations of cytokines that induce AD-related reactions with high efficiency. AD-HSE stimulated by the Th2 cytokines IL-4/IL-13 was shown to induce epidermal formation, altered gene expression and the secretion of several cytokines similar to AD skin.

First, IL-4/IL-13-stimulated AD-HSE, like other in vitro AD-like models and AD patients, had altered epidermal morphology resembling spongiosis or hyperkeratosis. IL-4 and IL-13 have been reported to induce spongiosis in epidermal models [33]. The morphology of the epidermis was deformed in our study in a concentration-dependent manner, similar to that in AD patients; when stimulated for 14 days, a loose keratin morphology, including desmosome changes in the SC, was observed. This result was consistent with the finding that IL-4 reduces the cohesive force of the SC and the expression of DSG1 in differentiated human cells [34]. DSG1 expression reduced by IL-4 suppresses desmosome maturation and affects the formation of keratinocytes, during which keratinocytes are less likely to bind and the barrier weakens. As a result, desquamation by IL-4/IL-13 is promoted in SCs [11]. The proper aggregation of the SC appears to be compromised in AD-HSE.

Second, in order to investigate changes in the epidermal barrier function of AD-HSE, the expression of barrier function proteins was examined by IHC staining. During the process of epidermal formation, stratum basale NHEKs expressing KRT5 and KRT14 divide and migrate to the SS layer, after which corneocytes convert the keratin profile to KRT1/KRT10. Keratin binds to the desmosome, a specific cell–cell adhesion structure [35,36]. Cornified envelope precursors, including IVL and transglutaminase-1 (TGM1), exist in the SS [36]. In the next layer, SG, keratohyalin cells form keratohyalin granules consisting primarily of profilaggrin and LOR [37]. In this study, there was no difference in the expression of epidermal proteins KRT14 and KRT10 in AD-HSE stimulated with IL-4/IL-13 for 3 days, whereas AD-HSE stimulated for 7 and 14 days showed reduced expression of KRT14 and KRT10, which was attributed to reduced KC differentiation. It can be inferred that the differentiation of SG and SC occurred from the seventh day onward.

We confirmed that IL-4/IL-13-induced AD-HSE had reduced expression of FLG, LOR and IVL proteins associated with barrier formation and function. This is because the Th2 cytokines IL-4/IL-13 use JAK to initiate signaling and activate STAT6 and STAT3 to strongly inhibit the expression of EDC molecules, including FLG, LOR and IVL, in NHEKs [12,14]. Downregulation of these barrier-related proteins in AD-HSE indicated impaired barrier function. Decreased expression of IVL inhibits the formation of a highly differentiated nucleus-free cell, the coned envelope [11]. The skin barrier function of the coned envelope, which forms an insoluble and hard structure in the SC layer of AD-HSE, prevents water loss and protects the skin from harmful external environmental influences, was compromised in our model. FLG is known to play an important role in the pathophysiology of AD because it is involved at various levels in the formation and maintenance of the correct epidermal barrier [38], partly owing to its alteration of keratin filament aggregation and functional corneocytes [39]. FLG alteration affects natural kinematic factor levels, which in turn can alter skin hydration and pH values [40,41]. Dehydration of the skin weakens the epidermal barrier and causes itching. Elevated skin surface pH improves the activity of proteases responsible for the exfoliation of keratin, but reduces the activity of enzymes involved in barrier lipid synthesis. Such FLG alteration weakens the epidermal barrier, enhances the penetration of allergens and pathogens and can cause skin inflammation [42]. Patients with AD also show alteration of the epidermal barrier regardless of *FLG* genotype, and strong reductions in *FLG* expression levels may be observed in AD skin (lesions and non-lesions) [43]. Downregulation of FLG in the epidermis can impair barrier permeability and, as a result, cause inflammation, especially at the lesion site [27,42]. We confirmed by SEM that the surface of the epidermis was roughened by cytokines and the shape of the surface was wrinkled. Surface changes appeared to be due to dehydration, increased pH and exfoliation activity of keratin due to decreased expression of barrier-related proteins such as FLG, LOR and IVL. A change in the shape of the SC layer of AD-HSE was observed in the SEM images of the SC cross-section. This appears to be typical spongiosis and hyperkeratosis, in which IL-4/IL-13 cytokines suppress the maturation of desmosomes and weaken the binding of corneocytes. As a result, desquamation accelerated and aggregation of the SC was weakened. In this study, we confirmed that IL-4/IL-13-induced AD-HSE had inhibited barrier formation in the same manner as AD and reproduced the inflammatory state.

Third, the Th2 cytokines IL-4 and IL-13, through IL-4Ra signaling, activated STAT6 signaling and activators, leading to a decrease in *FLG, LOR* and *IVL* gene expression. Expression was reduced not only at the protein level but also at the gene level. To evaluate the AD phenotype in AD-HSE by IL-4/IL-13, we assessed the expression of the AD-related gene *CAII*. According to in vitro studies, *CAII* is involved in cell pH, water transport and maintenance of ionic homeostasis [44]. *CAII* levels are known to be elevated by Th2 cytokines, especially in AD lesions [45]. The AD-HSE produced showed increased *CAII* gene expression. These results indicate that our AD-HSE model reconstituted AD-like inflammatory conditions in in vitro experiments.

Fourth, treatment of AD-HSE with IL-4/IL-13 induced secretion of TNF-α, IFN-γ, GM-CSF, IL-1α, IL-8, IP-10, RANTES, VEGF, EGF, resistin, IL-12, IL-13, PDGF-BB, PIGF-1, β-NGF, MCP-1, MIP-1α, IL-4, IL-10, FGFb, leptin, IGF-1, TGF-β, Adipo, IL-17A and IL-1β. Each of these cytokines is known to be involved in AD or inflammatory diseases [46,47,48,49,50,51]. AD is characterized by the aberrant expression of various inflammatory factors regulated by Th1 and Th2 cells [52]. In the chronic phase of AD, Th1 cytokines such as IL-12 and IFNγ also increase [30,53], and the expression of MIP-1a is characteristic of acute AD [54]. TGF-β is essential for the regulation of allergic diseases, including AD [55]. Leptin promotes IL-1, IL-6 and TNF-α production through its proinflammatory activity, which also affects psoriasis [56]. Thus, our model successfully recapitulates AD. 

The expression of IL-8 and IP-10 in NHEKs is associated with psoriasis [57,58]. High expression of RANTES, MCP-1 (CCL2) and IL-2R appears in the epidermis of patients with AD and in psoriasis patients [59,60,61]. In cutaneous pathologies characterized by increased angiogenesis, such as psoriasis, VEGF and PlGF-1 expression is increased [62,63]. β-NGF is increased in psoriatic skin [64], and more NGF is produced in psoriatic NHEKs [65]. EGF receptors (EGFRs) have been found to be upregulated in chronic inflammatory skin diseases such as psoriasis, atopic dermatitis and allergic contact dermatitis [66]. The increased expression of IL-8, IP-10, RANTES, MCP-1 (CCL2), IL-2R, VEGF, PlGF-1 and β-NGF indicates that our AD-HSE model can trigger a psoriatic response.

Resistin activates inflammatory cytokines, including TNF-α, IL-1β, IL-6 and IL-12, via the NF-κB signaling pathway [67]. PDGF-BB is known to affect re-epithelialization and tissue remodeling [68,69] and is related to the paracrine effects of HDFs on NHEKs [70]. FGFb is involved in several inflammation-related diseases and is highly expressed in HDFs [51,71]. HDFs secrete IGF-1 and support the proliferation of NHEKs in the epidermis through the activation of IGF-1R. The IGF-1/IGF1R pathway is associated with numerous hyperplastic epidermal disorders, such as psoriasis [72,73]. Secretion of cytokines such as resistin, PDGF-BB, FGFb and IGF-1 is increased due to the inflammatory response and overexpression by corneocytes. G-CSF accelerates the healing of exfoliated skin after irradiation [74]. High expression of G-CSF in all experimental groups appeared to be a feedback response for skin healing. In particular, the expression of 15 ng/mL IL-4/IL-13 for 14 days promoted recovery after the vigorous exfoliation of the epidermis. The increased expression of each cytokine indicated that AD-HSE can reproduce the cytokine profile of AD skin. The AD-HSE described in our study can be used to mimic the important physiological features of atopic dermatitis, and thus it could be used as a potent tool for studying AD. In contrast to other models that depend on mice, our AD-HSE better models human physiology and could be useful for future research on the development of therapeutic agents for atopic dermatitis.

## 4. Materials and Methods 

### 4.1. Fabrication of Pumpless Microfluidic Chip and Microphysiological System

The manufacturing process for a pumpless microfluidic SOC was described in a previous paper [25]. The pumpless skin-on-a-chip consists of a glass slide, a lower polydimethylsiloxane (PDMS) chip layer, an upper PDMS chip layer and a 0.4 μm porous membrane (It4ip, Wallonia, Belgium) between the two chip layers. Each configuration was O_2_ plasma bonded with a CUTE-1MP product (Femto Science Inc., Hwaseong, Korea), and the layers were bonded sufficiently strongly by surface modification. The upper PDMS chip was designed to have a cylindrical culture chamber with a diameter of 8 mm in the center of the chip, and culture solution storage chambers were arranged on both sides and connected via a lower channel. The culture medium was designed to be perfused through the microfluidic channel of the lower chip and supplied to the 3D skin equivalent through the polyester membrane of the culture chamber (Figure 7a). We used a gravity flow system to create a 3D culture in a fluid environment. The system consisted of a driving PC, a motor and a dish holder that can transmit movement to the chip, and the desired tilt angle and rotation frequency can be controlled through another program. After a set time lapse, the culture medium flowed in the opposite direction while tilting in the opposite direction, and the medium was effectively recirculated. The flow rate of the culture medium in the microfluidic channel could be controlled primarily by adjusting the tilting angle, with a volumetric flow rate of 10 μL/min at a tilting angle of 15° [75] (Figure 7b).

### 4.2. Surface Functionalization

In our study, a crosslinker was applied to the wall surface in contact with collagen to suppress the contraction of the collagen substrate. The experiment was conducted by coating PDMS with sulfosuccinimidyl 6-(4′-azido-2′-nitrophenylamino)hexanoate (Sulfo-SANPAH), a photocrosslinking agent, and fibronectin, and attaching a rat tail collagen matrix mixed with HDFs.

Sulfo-SANPAH (ProteoChem, Hurricane, UT, USA) is a heterobifunctional crosslinker containing an amine-reactive NHS ester and a photoactivated nitrophenyl azide group. For the PDMS surface sensitization procedure, Sulfo-SANPAH was dissolved in deionized water at a working concentration of 10 mM. The PDMS surface was completely covered with a Sulfo-SANPAH solution and exposed to UV light with a light output of 30 mW/cm. Sulfo-SANPAH was crosslinked with a double bond on the surface of PDMS via a group of nitrophenyl azides during UV treatment. After UV exposure, the PDMS chamber was washed with phosphate-buffered saline (PBS). The fibronectin solution (66 μg/mL) was treated for at least 3 h, and the treated surface was considered functionalized. A collagen solution containing cells was added to the PDMS chamber immediately after surface functionalization. When the collagen solution came into contact with the Sulfo-SANPAH-treated surface and gels, the collagen fibers were crosslinked to the PDMS surface via the open NHS ester (Figure 8).

### 4.3. Cell Culture

HDFs (Bio Solution Co. Ltd., Seoul, Korea) and NHEKs (Bio Solution Co. Ltd.) were cultured in an incubator humidified with 5% CO_2_ at 37 °C. HDFs were cultured using FGM-2 HDF growth medium (Lonza, Basel, Switzerland), and NHEKs were cultured in KGM-Gold NHEK Growth Medium (Lonza). Both primary cells were used at passages 4–6. 

### 4.4. Construction of Atopic Dermatitis Skin Equivalent Model

Rat tail collagen type I (Corning), 10× DMEM medium, 0.5 N NaOH, HDF suspension (final cell concentration, 5.0 × 10^5^ cell/mL) and media were mixed to neutralize the gel. The rat tail collagen concentration was set to 6.12 mg/mL. To fabricate the dermal layer (DL), the collagen–HDF suspension was seeded on the chip to a height of 3 mm and deposited for 40 min in a 37 °C incubator with 5% CO_2_. Thereafter, the DL was cultured with FGM-2 HDF growth medium for 5 days, and the medium was changed every day. After that, NHEKs were seeded on the DL (1.0 × 10^6^~5.0 × 10^6^ cell/cm^2^) and cocultured for 2 days. KGM-Gold Keratinocyte Growth Medium was supplied only above the DL-NHEKs, and FMG-2 media was supplied to the channel of the chip. This medium supplies E-media that induces the differentiation of NHEKs for 3 to 7 days and at the same time provides an environment similar to that of real skin by exposing it to air (E-media composition: DMEM/Ham’s F12, 10 ng/mL EGF-1, 0.4 μg/mL hydrocortisone, 5 μg/mL insulin, 5 μg/mL transferrin, 2 × 10^−11^ M 3,3,5-triiodo-L-thyonine sodium salt, 10-10 M cholera toxin, 10% (*v*/*v*) FBS, 1% penicillin/streptomycin). Recombinant human IL-4 protein and recombinant human IL-13 protein (R&D system, Minneapolis) treatment was performed at the air exposure stage. IL-4 and IL-13 were added in the same concentration. Experiments were performed at concentrations of 0, 5, 10 and 15 ng/mL. All cultures were incubated at 37 °C with 5% CO_2_ (Figure 9).

### 4.5. Measurement of Contraction of Human Skin Equivalents

The formation process of 3D culture HSE seeded in the culture chamber of a pumpless microfluidic chip was photographed using a digital microscope (Dino-Lite, Hsinchu, Taiwan) to measure the degree of contraction of the entire tissue area. The cultured tissue was observed every 24 h, and the area was measured using the ImageJ Fiji program. Four samples were measured; the mean value, standard deviation, and *p*-value for the difference in contraction for each sample were calculated using the Prism program, and the data were graphed.

### 4.6. Hematoxylin and Eosin (H&E) Staining

The sample was fixed in 10% formalin solution and then treated with paraffin wax. Tissues containing paraffin were cut to a thickness of 4 μm using a microtome. Sectioned tissue (4 μm) was attached to a glass slide and subjected to H&E staining or IHC. Sections were stained with hematoxylin solution (Sigma-Aldrich Co. Ltd., Lexington, MA, USA). After washing with water, the sections were stained with eosin solution (Sigma-Aldrich Co. Ltd.). The stained tissue was observed using an inverted optical microscope (Olympus, Tokyo, Japan). 

### 4.7. Immunohistochemistry Staining (IHC) Staining

For IHC staining, paraffin-embedded tissue sections were cultured with specific primary antibodies: cytokeratin 10 (Abcam, rabbit monoclonal, human (EP1607IHCY), ab76318), cytokeratin 16 (rabbit monoclonal (EP1615Y), human, ab76416), filaggrin (rabbit polyclonal, human, ab81468), loricrin (rabbit polyclonal, human, ab85679) and involucrin (rabbit polyclonal, human, ab53112). Secondary goat anti-rabbit IgG H&L (HRP) (ab205718) antibody was used at a 1:200 dilution. 3,3-Diaminobenzidine (DAB) was used to detect the target protein. The slides were visualized with an optical microscope (Olympus, Tokyo, Japan) equipped with an optical camera (DP73-ST-SET, Olympus, Tokyo, Japan). The results of five random shots of each sample were subjected to IHC image analysis using the software ImageJ Fiji program. The measurements were analyzed using Prism software.

### 4.8. RNA Extraction from Human Skin Equivalent

One milliliter of TRIzol Reagent (Invitrogen, Waltham, MA, USA) was added to the cultured SE tissue. Two hundred microliters of chloroform was added, and the mixture was homogenized and then centrifuged at 12,000× *g* for 15 min at 4 °C to obtain phase separation. The upper layer (liquid phase) of the three separated layers was transferred to a fresh tube. RNA was precipitated by adding 0.5 mL isopropanol and incubating on ice for 10 min, then centrifuging for 10 min at 12,000× *g* at 4 °C. The supernatant was removed, and the pellet was washed with 1 mL of 70% ethanol with DEPC-treated water and then centrifuged at 7500× *g* for 10 min at 4 °C. The supernatant was removed and the RNA pellet was dried and dissolved in an appropriate volume of DEPC-treated water. The quality and quantity of RNA were checked using a NanoDrop spectrophotometer (SpectraMax M2, Molecular Devices, LLC, San Jose, CA, USA).

### 4.9. Quantitative Real-Time PCR

Two-step RT-qPCR was performed by synthesizing cDNA from total RNA using amfiRivert cDNA Synthesis Platinum Master Mix (GenDEPOT, Katy, TX, USA). The qPCR was performed using this cDNA with LightCycler 480 SYBR Green I Master (Roche, Basel, Switzerland) from LightCycler 480 Instrument II (Roche, Basel, Switzerland). GAPDH (glyceraldehyde-3-phosphate dehydrogenase) was used as a housekeeping gene. Values were calculated using the delta-delta CT method. The primers used for qPCR are listed in Table 1.

### 4.10. Analysis of mRNA Expression by RT-PCR and Gel Electrophoresis

The synthesized cDNA was used to amplify the desired target genes *GAPDH*, *FLG*, *IVL*, *LOR* and *CAII*, using the AccuPower PCR PreMix & Master Mix (Bioneer, Daejeon, Korea). The primers used are listed in Table 1. A DNA ladder (Bioneer) and the DNA sample to be separated were loaded on a 1.5% agarose gel and electrophoresis was performed at 100 V. The bands separated by electrophoresis were photographed using the Gel Doc System. 

### 4.11. Scanning Electron Microscopy

HSE was fixed overnight with 2.5% glutaraldehyde followed by 1% osmium tetroxide. A series of ethanol concentrations (50% to 100%) was used for dehydrating samples for 15 min each. In the final step, the sheet was treated with 100% isoamyl acetate and dried using a critical point dryer. The sample was then sputter-coated with Au before being imaged using a scanning electron microscope (JEOL Ltd., Tokyo, Japan).

### 4.12. Enzyme-Linked Immunosorbent Assay Measurement

Culture medium was collected from both medium chambers and lower channels of SOC. The Human Cytokine ELISA Plate Array I Kit (Signosis, Inc., Santa Clara, CA, USA) was used to monitor the binding of 32 human cytokines. First, 100 μL of the collected medium was added to each well and reacted for 2 h, and then washed with an assay wash buffer. Then, 100 μL of the biotin-labeled antigen mixture was added to each well, which was shaken tightly for an hour. After washing with the assay wash buffer, 100 μL of streptavidin-HRP conjugate was added, followed by mixing for 45 min. Washing with the assay wash buffer was then followed by incubation with a 100 μL substrate for 40 min. Then, 50 μL stop solution was placed in each well. The optical density of each well was determined with a microplate reader at 450 nm.

### 4.13. Statistical Analysis

Data from at least three independent experiments are expressed as the mean ± standard deviation. Statistical analysis was performed using Prism software 9.3.0 version (GraphPad Software Inc., Santa Clara, CA, USA). Statistical significance was determined using two-way ANOVA.

## 5. Conclusions

Currently, the etiopathogenesis of atopic dermatitis is not well understood because there is no physiologically appropriate disease model in terms of disease complexity and multifactoriality. In addition, treatment of AD currently requires a viable in vitro model to test new molecules. A microphysiological system technology has the advantage of creating an environment that is similar to complex in vivo conditions. SOC utilizes HSE, which has the same characteristics as human skin, as it allows evaluating skin toxicity for new molecules and raw materials to be performed quickly and easily. This study developed full-thickness HSE from a pumpless SOC stimulated with IL-4 and IL-13 to induce AD. AD-HSE can be used to mimic the key pathological features of atopic dermatitis, overcoming the limitations of existing studies that rely solely on mouse models and have been unable to translate their effects to humans. Our results will be useful for future research on the development of therapeutic agents for atopic dermatitis.

## Figures and Tables

**Figure 1 ijms-23-02116-f001:**
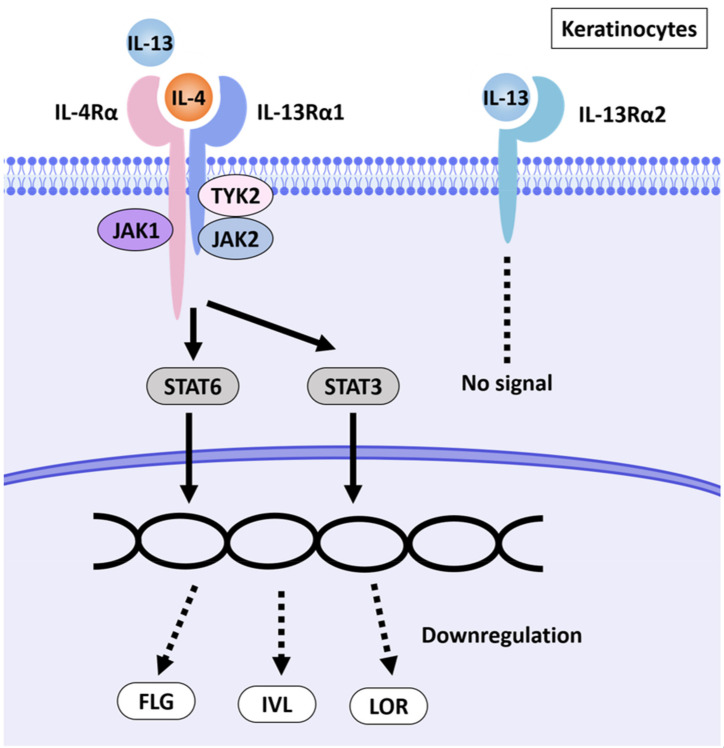
Schematic illustration of AD pathways induced by IL-4 and IL-13. IL-4 and IL-13 have partly shared receptor systems. Corneocytes express the IL-4Rα/IL-13Rα1 complex. IL-4 and IL-13 share IL-4Rα/IL-13Rα1 in NHEKs, activate the JAK1/JAK2/TYK2-STAT6 and -STAT3 pathways and inhibit the expression of EDC molecules such as FLG, LOR and IVL. NHEKs also express IL-13Rα2, a prey receptor for IL-13. IL-13Rα2 binds to IL-13 with high affinity but does not transmit a signal [13,14,15].

**Figure 2 ijms-23-02116-f002:**
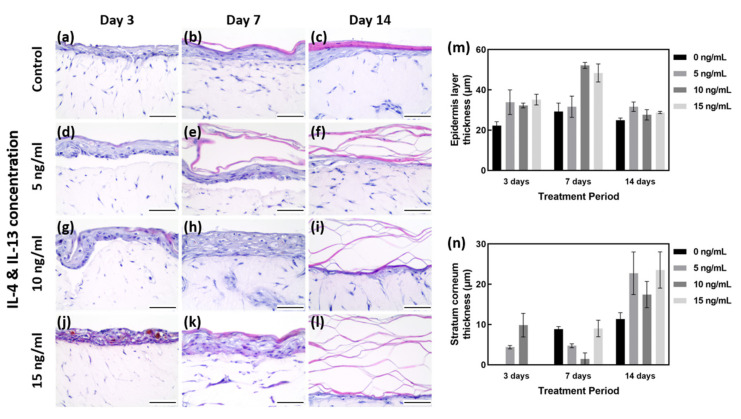
H&E staining of AD-HSE. (**a**–**c**) H&E staining of HSE cultured in general medium (without IL-4/IL-13 stimulation) for 3, 7 and 14 days as a baseline control. (**d**–**f**) H&E staining of AD-HSE stimulated with 5 ng/mL IL-4/IL-13 for 3, 7 and 14 days. (**g**–**i**) H&E staining of AD-HSE stimulated with 10 ng/mL IL-4/IL-13 for 3, 7 and 14 days. (**j**–**l**) H&E results of AD-HSE stimulated with 15 ng/mL IL-4/IL-13 for 3, 7 and 14 days. Scale bar is 50 μm. (**m**) Graph quantified by measuring the thickness of all epidermis layers, including stratum corneum. (**n**) Graph quantified by measuring only the thickness of stratum corneum.

**Figure 3 ijms-23-02116-f003:**
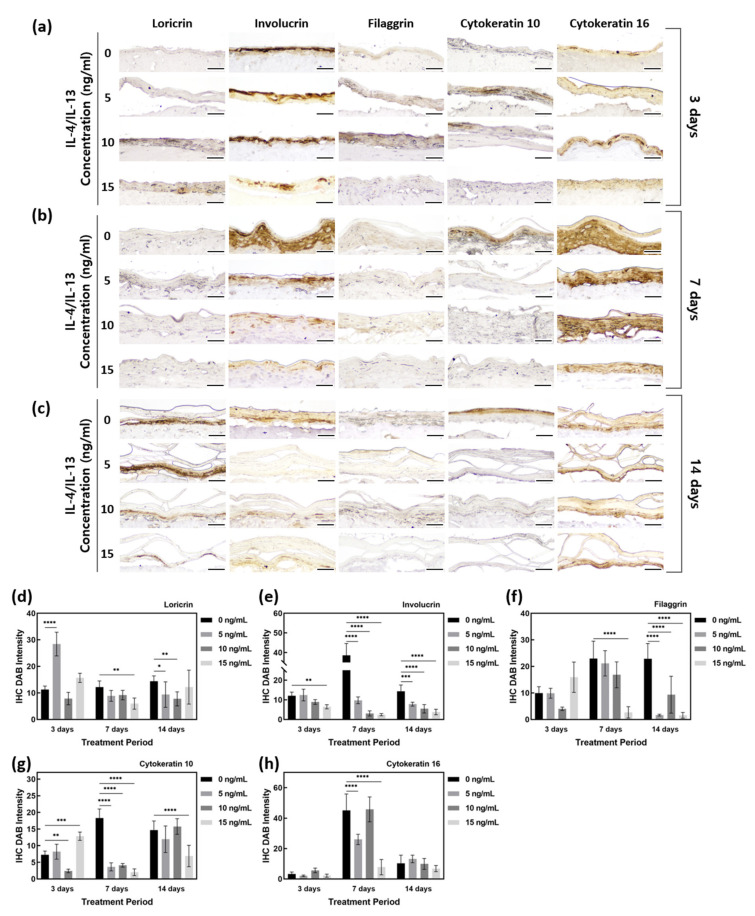
Quantification of IHC staining of AD-HSE stimulated with IL-4/IL-13. (**a**) IHC-stained images of epidermis-forming proteins of AD-HSE stimulated with IL-4/IL-13 for 3 days. (**b**) IHC-stained images of epidermis-forming proteins of AD-HSE stimulated with IL-4/IL-13 for 7 days. (**c**) IHC-stained images of epidermis-forming proteins of AD-HSE stimulated with IL-4/IL-13 for 14 days. (The scale bar sizes of a, b and c are 50 μm.) (**d**–**h**) Epidermis-forming proteins loricrin, involucrin, filaggrin, cytokeratin 10 and cytokeratin 16 (*n* = 5; *, *p* < 0.05; **, *p* < 0.01; ***, *p* < 0.001; ****, *p* < 0.0001).

**Figure 4 ijms-23-02116-f004:**
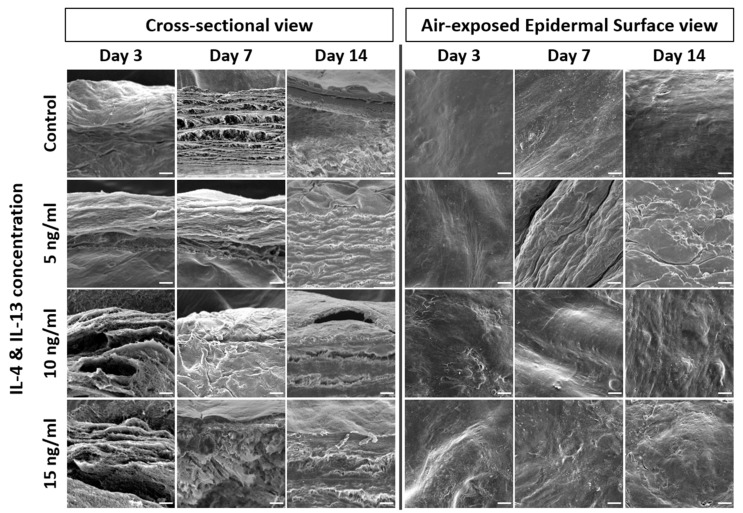
SEM of AD-HSE. SEM of cross-sectional and air-exposed epidermal surface views of AD-HSE stimulated with IL-4/IL-13 for 3, 7 and 14 days. Scale bar is 20 μm.

**Figure 5 ijms-23-02116-f005:**
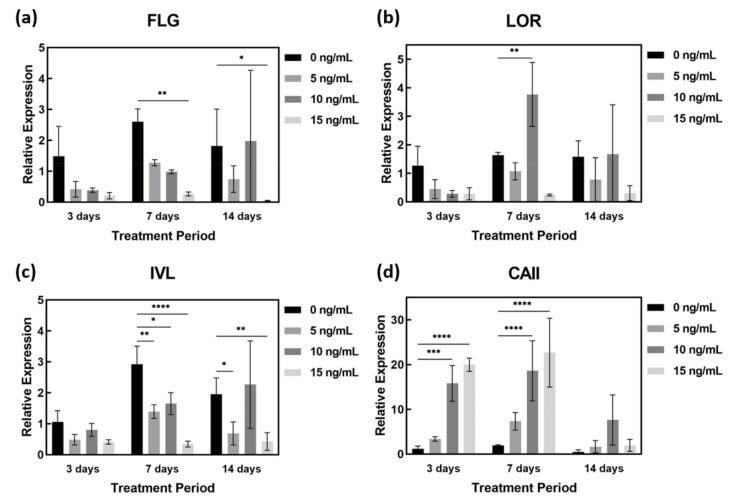
qPCR of IL-4- and IL-13-stimulated AD-HSE. AD-HSE treated with IL-4/IL-13 by concentration and period and controls: (**a**) *FLG*, (**b**) *LOR*, (**c**) *IVL*, (**d**) *CAII* (*n* = 3; *, *p* < 0.05; **, *p* < 0.01; ***, *p* < 0.001; ****, *p* < 0.0001).

**Figure 6 ijms-23-02116-f006:**
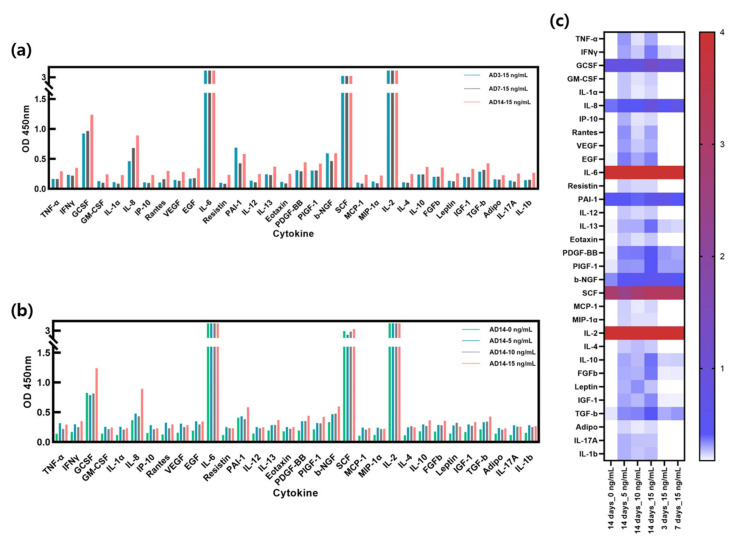
Cytokine expression in AD-HSE. (**a**) ELISA results with human cytokine plate array of AD-HSE stimulated with 15 ng/mL IL-4/IL-13 for 3, 7 and 14 days each. Most cytokines were highly expressed when AD-HSE was stimulated for 14 days. (**b**) ELISA results of AD-HSE stimulated with 0, 5, 10 and 15 ng/mL IL-4/IL-13 for 14 days, showing differential expression of cytokines depending on IL-4/IL-3 concentration. (**c**) The results of ELISAs shown by a heat map.

**Figure 7 ijms-23-02116-f007:**
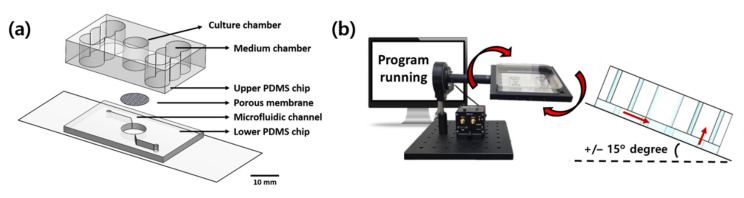
Schematic diagram of the pumpless microfluidic chip and gravity flow system in action. (**a**) Configuration diagram of pumpless microfluidic chip. (**b**) Schematic diagram of the operation of the gravity flow system. Both sides are shaken at a 15° slope to allow the medium to circulate through the microfluidic channels.

**Figure 8 ijms-23-02116-f008:**
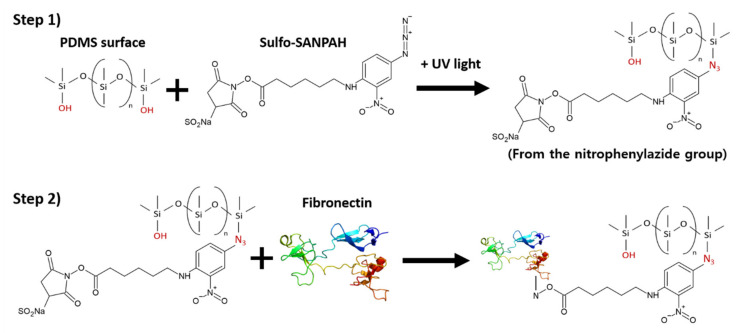
Chemical structures of the crosslinking agent Sulfo-SANPAH and the actual appearance of AD-HSE. Chemical reaction scheme using Sulfo-SANPAH. Sulfo-SANPAH is a heterobifunctional crosslinker containing an amine-reactive NHS ester and a photoactivatable nitrophenyl azide group. Crosslinking agent treatment sequence: (**Step 1**) Sulfo-SANPAH is crosslinked to PDMS through a nitrophenyl azide group, and nitrene with high reactivity is formed from the nitrophenyl azide group during UV treatment, which is crosslinked by a double bond on the PDMS surface. (**Step 2**) The UV-light-processed Sulfo-SANPAH is treated with fibronectin. After crosslinking of nitrene to the PDMS surface, binding to fibronectin forms a stronger crosslink. When the collagen solution comes into contact with the Sulfo-SANPAH-treated surface and gels, the collagen fibers are crosslinked on the PDMS surface through an open NHS ester.

**Figure 9 ijms-23-02116-f009:**
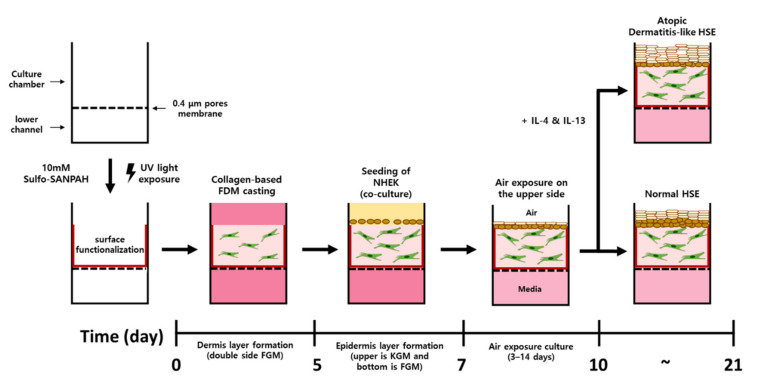
Schematic diagram of the AD-HSE formation process in the microphysiological system. The culture chamber of the pumpless SOC is shown for different incubation periods. The surface of the culture chamber is functionalized with Sulfo-SANPAH, and a dermal layer is formed for 5 days. On the 7th day after the start of culture, skin equivalents are formed through exposure to air and treated with the Th2 cytokines IL-4/IL-13.

**Table 1 ijms-23-02116-t001:** Sequences of forward and reverse primers used for qPCR analysis.

Gene	Forward Primer (5′ to 3′)	Reverse Primer (5′ to 3′)
*GAPDH*	5′-CTCCTCTGACTTCAACAGCG-3′	5′-GCCAAATTCGTTGTCATACCAG-3′
*FLG*	5′-GGAGTCACGTGGCAGTCCTCACA-3′	5′-GGTGTCTAAACCCGGATTCACC-3′
*IVL*	5′-CCGCAAATGAAACAGCCAACTCC-3′	5′-GGATTCCTCATGCTGTTCCCAG-3′
*LOR*	5′-CCGGAGATGGTGGCCTTCTCTCT-3′	5′-GGCCTGATGTGAGTTGCCATGCT-3′
*CA II*	5′-CAAGAGAGCTGAAGACTATCCCA-3′	5′-TGAAGTCCGAAGTAATCCTCCT-3′

## Data Availability

The data presented in this study are available on request from the corresponding author.

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
