# Peer review of "An Interleukin-4 and Interleukin-13 Induced Atopic Dermatitis Human Skin Equivalent Model by a Skin-On-A-Chip"

_ijms, 2022, doi:10.3390/ijms23042116_

Round 1

Reviewer 1 Report

This is an interesting article about a new AD animal model. Nevertheless, the article needs to be reorganized and it would benefit from English review.

The introduction should be reorganized. There are information repeated in several paragraphs.

Page 1. Line 17. Delete “were observed”.

Page 2. Line 34. Include the reference after “cell differentiation”. Rephrase this sentence: Stratum basal expansion, stratum spinosum and stratum granulosum (SG) reduction and stratum corneum (SC) thickening are observed because of terminal keratin cell differentiation defects.

Line 37. It is better to say that the etiopathogenesis of AD is not completely understood instead of the mechanism of progression. Rephrase:  “….understood, and there is no AD appropriate AD disease models”.

Line 40. Omit “each inducind the other in a cicious circle”

Line 39-42. You are repeating information that you previously mention at the paragraph beginning. Please join the information together at the beginning or the end.

Line 45. Instead of “such as” use including

Line 50. There is no need that you mention the layer where the process happens.

Line 51. Change by “, Figure 1”

Line 62-69. You mentioned in the previous paragraph that there is a need to develop AD skin models. There is no need that you mention it previously.

Line 64. It would be better to employ another word instead of “resistance”. Pleas rephrase line 63-63

Line 68. Are ECM scaffolds being investigated for AD? Please specify

Line 71. Add the reference at the end of the sentence

Line 73. To instead of “in order to”

Line 73. Add “as” drug screening …

Line 74-75. Delete “in this study” and add the references at the end of the sentence

Line 78-81. Is part of the material and methods section

Line 82. Add the aim of this study was 1) to develop…. 2) to investigate the morphological properties…. 3) to evaluate the induction of AD….

Line 86-87. Delete “ we expect that….”

Material and methods. Please add more information about the statistical analysis. What p value do you consider statistical significative?

Please explain all the acronyms the first time they appear, for example PDMS

Results

The results section should only include the results that you found in your study. You should not include information comparing with other studies or talking about the way you did your experiments (material and methods)

Line 90-92. They are not results

Line 92-96. They are part of the material and methods.

Line 98-103. Here you are mixing sentences that should be included in the introduction an in the material and methods section, but they are not results.

Line 105. How can your say HSE are well formed? Please add more information about the characteristic in HE.

I think you could add statistical analysis and objective parameters regarding the thickness of the strata, the number of collagen fibers and other histological data comparing the different types of HSE

Line 131-133. They are not results that you find in your study.

Line 135-152. Please add numeric data and p values in the text.

Line 144-145 are not results

Line 160.164 are not results.

Line 168-170. Add numeric data

2.4-2.5 Add numeric data and p values please

Line 175-176 is part of material and methods

Line 183-186 are part of the discussion

Line 192-196 are not results.

Line 196-198 are part of material and methods

Line 223-25. Delete them

Conclusion should summarize the main characteristic found in the AD-HSE models.

Line 499-500. “Currently, the mechanism of progression of atopic dermatitis is not well understood 499 because there is no physiologically appropriate disease model in terms of disease com-500 plexity and multifactoriality.” This is not true because there are more reasons because the etiopathogenesis of AD is not well understood.

Line 502, 504. Use only the acronym when you have previously defined the word.

Line 502-506. It is a repeated sentence of the introduction

You should also be consistent with the acronyms. They should be defined the first time they appear and after that you should always use the acronym instead of all the words.

Author Response

Reply to Reviewer 1

We appreciate the thorough review of our manuscript. We revised the manuscript extensively for clarity according to the Reviewer's comments

Comments and Suggestions for Authors

This is an interesting article about a new AD animal model. Nevertheless, the article needs to be reorganized and it would benefit from English review.

  1. The introduction should be reorganized. There are information repeated in several paragraphs.

Answer: The structure of the contents has been changed according to the items pointed out below.

  1. Page 1. Line 17. Delete “were observed”.

Answer: It has been deleted.

  1. Page 2. Line 34. Include the reference after “cell differentiation”. Rephrase this sentence: Stratum basal expansion, stratum spinosum and stratum granulosum (SG) reduction and stratum corneum (SC) thickening are observed becaus All references in the sentence have been changed to the end of the sentence.e of terminal keratin cell differentiation defects.

Answer: “Due to widespread defects in terminal keratin cell differentiation, there is an expansion of the stratum basale, as well as a reduction of the stratum spinosum and stratum granulosum (SG) and thickening of the stratum corneum (SC).” We deleted the foreign text and changed it like this: “Stratum basal expansion, stratum spinosum and stratum granulosum (SG) reduction and stratum corneum (SC) thickening are observed because of terminal keratin cell differentiation defects.”
All references in the sentence have been changed to the end of the sentence.

  1. Line 37. It is better to say that the etiopathogenesis of AD is not completely understood instead of the mechanism of progression. Rephrase:  “….understood, and there is no AD appropriate AD disease models”.

Answer: Depending on what was pointed out, we changed to “the etiopathogenesis of AD” instead of “the mechanism of AD progression”. The changed text is as follows:
Currently, the etiopathogenesis of AD is not well understood, at least partially because there is no physiologically appropriate disease model in terms of disease complexity and multifactorial nature.

  1. Line 40. Omit “each inducind the other in a cicious circle”

Answer: “each inducing the other in a vicious circle” Delete that part.

  1. Line 39-42. You are repeating information that you previously mention at the paragraph beginning. Please join the information together at the beginning or the end.

Answer: Since the beginning of the introduction and the information were repeated, We moved the sentence to the first part to connect to the contents of the first line.

  1. Line 45. Instead of “such as” use including

Answer: Changed to "Including" instead of "such as".

  1. Line 50. There is no need that you mention the layer where the process happens.

Answer:  During the differentiation process, IL-4, IL-13 and IL-22 suppress the expression of involucrin in the stratum spinosum layer, loricrin in the SG layer and the formation of the cornified envelope. We removed "involucrin in the stratum spinosum layer" and "in the SG layer" from the above sentence. We changed it as follows.
During the differentiation process, IL-4, IL-13 and IL-22 suppress the expression of involucrin, loricrin and the formation of the cornified envelope.

  1. Line 51. Change by “, Figure 1”

Answer:  We corrected.

  1. Line 62-69. You mentioned in the previous paragraph that there is a need to develop AD skin models. There is no need that you mention it previously.

Answer: This is the message we want to convey in this paragraph, so we moved it to the end of the paragraph.

  1. Line 64. It would be better to employ another word instead of “resistance”. Pleas rephrase line 63-63

Answer: We changed to "Regulation" instead of the expression "resistance". Animal testing is currently banned in Europe. I wanted to express that animal experiments tend to decrease. As you pointed out, "resistance" does not seem to match.

  1. Line 68. Are ECM scaffolds being investigated for AD? Please specify

Answer: The ECM study here mentions that the 3D culture model will be investigated as an alternative to animal experiments in the previous text. ECM has not been studied separately for AD models.

  1. Line 71. Add the reference at the end of the sentence

Answer:  Since this statement is a description of Skin-on-a-chip, we have moved the SOC reference to the end of the statement.

  1. Line 73. To instead of “in order to”

Line 73. Add “as” drug screening …

Answer: “In order to evaluate skin toxicity for new molecules and raw materials;” We tried to refer to drug screening as an example of something like evaluate skin toxicity for new molecules and raw materials. We deleted it because it seems to be a duplicate expression.

  1. Line 74-75. Delete “in this study” and add the references at the end of the sentence

Answer: We removed that part and moved the reference to the end of the sentence.

  1. Line 78-81. Is part of the material and methods section.

Answer: According to your comment, that part overlaps with the material and methods section, so we removed the statement.

  1. Line 82. Add the aim of this study was 1) to develop…. 2) to investigate the morphological properties…. 3) to evaluate the induction of AD….

Answer:  “This study developed full-thickness HSE from pumpless SOC stimulated with IL-4 and IL-13 to induce AD. The morphological properties of the epidermis, cortical protein and gene expression and cytokine secretion were investigated. We evaluated the inducibility of AD by our skin substitutes and their potential as a drug test evaluation model for therapeutic agents for AD.”
We changed to one text explaining the purpose of this experiment as follows:
“The aim of this study was to develop full-thickness HSE from pumpless SOC stimulated with IL-4 and IL-13 to induce AD, to investigate the morphological properties and to evaluate the induction of AD by our skin substitutes and their potential as a drug test evaluation model for therapeutic agents for AD.”

  1. Line 86-87. Delete “ we expect that….”

Answer: We deleted it according to your comment.

  1. Material and methods. Please add more information about the statistical analysis. What p value do you consider statistical significative?

Answer:  The P value was determined using a two-way ANOVA. It has already been mentioned, so there is nothing to add. It seems that the content about the software version used is not enough, so we added the program version and company name.

  1. Please explain all the acronyms the first time they appear, for example PDMS

Answer: Thank you for your good point. The explanation of the abbreviation for PDMS was missing, so we added it as “Polydimethylsulfonic (PDMS)”

  1. The results section should only include the results that you found in your study. You should not include information comparing with other studies or talking about the way you did your experiments (material and methods)

Answer: We have included a quote to explain that the results of our experiments are consistent with previous reports. This seems to be consistent with what was mentioned in the discussion section. As you pointed out, we agree that the results should only include the results. We corrected everything as commented.

  1. Line 90-92. They are not results

Answer: We deleted it.

  1. Line 92-96. They are part of the material and methods.

Answer: This statement has been moved to material and methods 4.2.

  1. Line 98-103. Here you are mixing sentences that should be included in the introduction an in the material and methods section, but they are not results.

Answer: We deleted it.

  1. Line 105. How can your say HSE are well formed? Please add more information about the characteristic in HE. I think you could add statistical analysis and objective parameters regarding the thickness of the strata, the number of collagen fibers and other histological data comparing the different types of HSE

Answer:  The HSE produced in our study consists of dermis layer formed by the differentiation of fibroblasts into ECMs. When keratinocytes differentiate, the stratum basale, stratum spinosum, stratum granulosum, and stratum corneum layer that form epidermis are separated and formed. Layer formation may appear different if differentiation is not done correctly, or if there are some alteration in early differentiation. Therefore, the well-formed epidermis uniformly forms the stratum basale, stratum spinosum, stratum granulosum, and stratum corneum layer that maintain a moderate thickness. Depending on the skin growth cycle, dehorning and healing occur to maintain proper thickness at all times. The explanation of epidermis formation has already been explained in the introduction, so We did not write it in the results.
We added the sentence below to the result :
“In Figure 2m and n, the epidermis thickness on days 3, 7 and 14 days under the 0 ng/mL (control) condition forms 22.20 to 29.22 μm. The SC layer was shown to gradually increase with the culture period (0-11.34 μm). Among the controls cultured for 7 days, the total thickness of epidermis was measured to be 29.22 μm on average, and the SC layer was measured to be 8.83 μm on average. Although the epidermis layer except for the SC was maintained at a level similar to that of the 3 days, it can be considered as the most well-formed tissue because the SC layer was formed properly.”

We added a statistical analysis comparing the HSE of each condition. The thickness of the SC layer of the Epidermis layer and the entire epidermis was measured. Added graphs to compare them as m and n in Figure 2.

  1. Line 131-133. They are not results that you find in your study.

Answer: We deleted it.

  1. Line 135-152. Please add numeric data and p values in the text.

Answer: We added it.

  1. Line 144-145 are not results

Answer: We deleted it.

  1. Line 160.164 are not results.

Answer: We deleted it.

  1. Line 168-170. Add numeric data

Answer:  Since this part is the result of SEM images, we can only explain the result for the morphological features of the tissue. As for the result of quantifying the histological data, We think that the result of further correction in 2.1 is sufficient. The reason for taking the SEM was to observe the actual AD-HSE morphology in more detail from the tissue staining results, such as changes in pH, water loss, hyperkeratosis, and changes in the epidermis due to extreme exfoliation.

  1. 4-2.5 Add numeric data and p values please

Answer: We added numeric data and p-values in paragraph 2.4.
2.5 is the data obtained by relative comparison of the ELISA result values, and there is no significant difference value and there is no content to add the p-value.

Thank you again for your kindly and helpful review.

Reviewer 2 Report

An original paper about a full thickness human skin equivalents consisting of human-derived cells were fabricated from pumpless microfluidic chips and stimulated with IL-4 and IL-13, investigating all properties of this human skin equivalent; only minor requests:

In the statistical analysis subsection of material and methods, you have to specify the version, the maker, and location of the program you used; you have also to specify what p-value you considered significant.

line 40-42 "Atopic dermatitis is an imbalance in the Th2 immune response where there is an increase in the gene expression levels of major Th2 cytokines during the acute phase" this paragraph needs some references, such as: doi: 10.1111/exd.14276.

Also in the in production, a description about the clinical phenotypes of AD should be added; here an interesting article: doi: 10.18176/jiaci.0519.

Thank You

Author Response

We appreciate the thorough review of our manuscript. We revised the manuscript extensively for clarity according to the Reviewer's comments

Comments and Suggestions for Authors

An original paper about a full thickness human skin equivalents consisting of human-derived cells were fabricated from pumpless microfluidic chips and stimulated with IL-4 and IL-13, investigating all properties of this human skin equivalent; only minor requests:

  1. In the statistical analysis subsection of material and methods, you have to specify the version, the maker, and location of the program you used; you have also to specify what p-value you considered significant.

Answer: We added information about the program used (program version, company name, etc.). The range of p-value values is described in the result figure caption. (*, p <0.05; **, p <0.01; ***, p <0.001; ****, p <0.0001)

  1. line 40-42 "Atopic dermatitis is an imbalance in the Th2 immune response where there is an increase in the gene expression levels of major Th2 cytokines during the acute phase" this paragraph needs some references, such as: doi: 10.1111/exd.14276.

Also in the in production, a description about the clinical phenotypes of AD should be added; here an interesting article: doi: 10.18176/jiaci.0519.

Answer: Thank you for your comments. We added the references as you recommended, as Ref. 3 and 4 in the revised version. This sentence has been repositioned to the front of the introduction.

Thank You.
